# The Potential Relationship between Gastric and Small Intestinal-Derived Endotoxin on Serum Testosterone in Men

Laura N. Phan [1],*, Karen J. Murphy [2,3], Karma L. Pearce [3], Cuong D. Tran [4] and Kelton P. Tremellen [5,6],*

[1] School of Clinical and Health Sciences, University of South Australia, Adelaide 5001, Australia
[2] Alliance for Research in Exercise, Nutrition and Activity, University of South Australia, Adelaide 5001, Australia; karen.murphy@unisa.edu.au
[3] Clinical and Health Sciences, University of South Australia, 108 North Terrace, Adelaide 5001, Australia; karma.pearce@unisa.edu.au
[4] CSIRO Health and Biosecurity, Gate 13, Kintore Ave, Adelaide 5000, Australia; cuong.tran@csiro.au
[5] Department of Obstetrics Gynaecology and Reproductive Medicine, Flinders University, Bedford Park 5042, Australia
[6] Repromed, 180 Fullarton Rd., Dulwich 5065, Australia
\* Correspondence: lan.phan@mymail.unisa.edu.au (L.N.P.); kelton.tremellen@flinders.edu.au (K.P.T.)

**Abstract:** The association between *H. pylori* and small intestinal permeability (IP) on serum testosterone levels in men as mediated by metabolic endotoxemia remains unclear. We sought to explore relationships using correlational analysis between *H. pylori* IgG class antibody levels and small IP via dual sugar probe analysis on T levels in 50 male participants of reproductive age. Sleep quality, physical activity levels, and Irritable Bowel Syndrome (IBS) symptom severity were measured as potential confounders. Measures for *H. pylori* (antibodies) increased small IP (lactulose/rhamnose ratio), and hypogonadism (testosterone) did not exceed diagnostic cut-off values for respective pathologies. There was no correlation between lactulose/rhamnose e ratio and GI function markers, zonulin, *H. pylori*, and IBS questionnaire scores; inflammatory markers, high-sensitivity C-reactive Protein (hsCRP) and Lipopolysaccharide-Binding Protein (LBP); nor endocrine markers, testosterone, Luteinizing hormone (LH), and Follicle-stimulating hormone (FSH). There was a moderate inverse relationship revealed between IBS symptom severity and LBP (r = −0.457, p = 0.004); and hsCRP and testosterone (r = −0.398, p = 0.004). This was independent of physical activity level and sleep quality, but not BMI, which supports the existing link between adiposity, inflammation, and hypogonadism currently present in the literature.

**Keywords:** endotoxemia; *Helicobacter pylori*; hypogonadism; intestinal permeability; small intestinal bacterial overgrowth; inflammation; testosterone deficiency

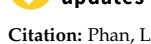



## 1. Introduction

In secondary hypogonadism (HG), failure of the testes to synthesise sufficient testosterone for spermatogenesis can lower the chances of healthy conception [1]. One clinical presentation of secondary HG is androgen deficiency, often leading to male infertility [2–5]. Male factor-infertility contributes to 25% of global infertility cases. In Australia, North America, and Eastern Europe, male-factor infertility rates are reported at 9%, 4–6% and 8–12%, respectively [6]. Secondary HG is associated with lifestyle factors traditionally associated with inflammation, including excess alcohol consumption and adiposity [7]. Exposure to endotoxins like Gram-negative bacterial lipopolysaccharide (LPS) have also shown involvement in androgen deficiency. Approximately 14% of male-factor infertility cases are associated with systemic inflammation [8]. This is supported by Boutagy et al.'s findings [9] which reveal that inflammatory cytokines present in semen and plasma, including Interleukin-6 (IL-6), IL-8, IL-10, and IL1-B, have been linked with decreased T production.

LPS is a glycolipid forming part of the Gram-negative bacterial wall and is a potent proinflammatory compound. LPS negatively affects testicular function both directly and indirectly. Leydig cells of the testis contain LPS-activated toll-like receptor 3 and 4 (TLR-3, TLR-4) and are, therefore, indicated in the expression of innate immune responses. As such, gonadal immune activation is consequently indicated in steroidogenesis disruption and reduction of T in multiple animal models. Isolated Leydig cells of mice and mallard ducks exposed to LPS injections demonstrated a significant decrease in steroidogenic acute regulatory (StAR) protein expression [10]. StAR activity initiates the conversion process between cholesterol to T from mitochondria, and is a likely mediator of T concentrations [11].

Direct mechanistic effects have also been observed. Male mice administered with LPS via injection have demonstrated significant decreases in T production and serum T in comparison to age and diet-controlled control mice [12]. Furthermore, experimental administration of LPS in animal models has demonstrated substantial suppression of leutenising hormone (LH) concentrations in sheep, cattle, non-human primates, and rats through the observed effect of decreased LH pulse frequency, suggesting a level of hypothalamic-pituitary dysfunction [11,13,14].

Supportive of animal findings, an observational study involving 75 healthy human males of reproductive age revealed a negative correlation between serum levels of lipopolysaccharide binding protein (LBP) levels, indicative of LPS exposure, inflammation [15], and T levels [16]. Similar biological effects, as mediated by inflammatory processes, have been demonstrated in experimental human models. Catheter administration of *Escherichia coli* (*E. coli*)-derived LPS (0.8 ng/kg body weight) in 17 healthy men resulted in profound increases in TNF-alpha (TNF-$\alpha$) and Interleukin-6 (IL-6), and significant reductions in T 6 h post-injection, in comparison to 16 men healthy men receiving intravenous placebo endotoxin. This effect was independent of gonadotrophin fluctuations to LH and follicular stimulating hormone (FSH) [16].

A potential underlying mechanism behind the LPS and T level correlations may be explained by metabolic endotoxemia (ME). ME is a state of low-grade, systemic inflammation induced by the passage of intestinal bacterial LPS into systemic circulation at sub-clinical levels (<200 pg/mL), where chronic activation of innate immune cell TLR-4 occurs. It is dissimilar to the relatively high median levels (300 pg/mL; 25–75% interquartile range, 110–726 pg/mL) found in patients at the onset of severe bacterial sepsis [17]. As LBP—an acute-phase reactant hepatically-produced in response to Gram-negative bacterial infections—binds to the Lipid A portion of the LPS molecule, complement is activated. This produces a pro-inflammatory immune response [9]. Elevations in serum LBP can be detected even in low LPS conditions, like ME. Therefore it is a suitable marker for ME as indicated in androgen deficiency [15].

Areas of Gram-negative bacterial colonisation are of concern, as LPS migrates from colonisation sites into circulation. The intestinal epithelium is a particularly fallible 'entry port' due to its naturally permeable nature, where mediation of epithelial tight junction regulation assists paracellular absorption and transcellular transport of macromolecules including bacterial by-products across intestinal capillaries [18]. Furthermore, the abundance of Gram-negative bacteria, even in the absence of active GI infection, further contributes to this dilemma. The average adult human intestinal microbiome—amassing 1.5 kg of bacteria, 70% of which is Gram-negative—contains the largest source of LPS in the human body. Approximately 1 g of unbound endotoxin is present in the average intestinal lumen at homeostasis [19–21].

Intestinal permeability (IP) is characterised by abnormal intestinal epithelial function which can lead to losses in immunological regulation and subsequent increases in 'non-discriminatory' circulatory migration of LPS. Conditions like Irritable Bowel Syndrome (IBS), which involve alterations to gut microbiome composition/ecology (dysbiosis), decreased immune defense of the mucosal immune system, and physical alteration of the mucosal barriers, can increase IP [22,23]. Crohn's Disease, where a degree of IP is expected, is also associated with poor T levels [24].

SIBO has also been shown to increase IP [25] and increase endotoxemia in patients with liver cirrhosis [26]. SIBO occurs when certain conditions, like proton pump inhibitor (PPI) use, increase intestinal pH levels. This causes an expansion in the number of pathobionts where the SI bacterial load exceeds the normal levels of $10^5$ colony-forming units (CFU)/mL of jejunal fluid [27]. Most bacteria are pH-sensitive and cannot thrive in acidic conditions. For this reason, relative to the more alkaline colonic environment, the SI has a low bacterial load ($10^4$ CFU/mL) compared to the colon at $10^{12}$ CFU/mL [16]. It is unknown whether SIBO-associated IP is a direct result of bacterial villous injury or an indirect result of immune activation, however [28]. Nevertheless, this suggests that SIBO is a plausible factor in inflammation-mediated testicular dysfunction despite current literature not yet highlighting a direct link between SIBO and testicular function.

Gastric permeability has also been shown to increase LBP. In particular, active CagA+ and VagA+ (cytotoxin-associated gene A [CagA]-and vacuolating cytotoxin A [VacA]-positive) *Helicobacter pylori* (*H. pylori*) virulent strain infections have been positively associated with serum LBP [29,30]. *H. pylori* infections cause gastric inflammation and subsequently gastric permeability, but not small IP, by activation of myosin light-chain kinase in epithelial cells to phosphorylate myosin light chain, and also by disrupting the tight-junctional proteins occludin, claudin-4, and claudin-5 [31,32]. *H. pylori* antibody levels have also been attributed to significantly decreased androgens and serum androstanediol glucuronide (3-$\alpha$-diol-G) (AAG) levels, an androgen biomarker [33].

Due to their involvement in increasing epithelial permeability at gastric and small-intestinal sites, respectively, and their relationship to increased LBP levels indicative of systemic inflammation, we believe *H. pylori* and IP are plausible factors in ME-mediated androgen deficiency.

## 2. Materials and Methods

This was a prospective, observational investigation in $n = 50$ men of reproductive age to determine whether *H. pylori* and IP could influence serum T levels through inflammatory influence. This study was conducted in accordance with the Declaration of Helsinki. Ethics approval for this study was obtained from the University of South Australia Human Research Ethics Committee (#202371). All participants provided written, informed consent prior to the commencement of data collection.

Inclusion criteria were male and between 18 and 50 years of age. Exclusion criteria were metabolic disorders (e.g., diabetes mellitus, metabolic liver disease), autoimmune disorders (e.g., Ehlers-Danlos syndromes, Graves' disease), infectious diseases (e.g., Hepatitis A, Hepatis B, HIV/AIDS), inflammatory bowel disease (IBD), pathological hyperlipidaemia (and lipid nephrosis or acute pancreatitis if accompanied by hyperlipidaemia), pulmonary disease, blood coagulation disorders, smokers, narcotic usage, excessive consumption of alcohol (>4 standard drinks per day), consumption of GI modulating supplements (e.g., probiotics, prebiotics, supplemental fibre), consumption of medication with immunosuppressive function (e.g., corticosteroids), consumption of antibiotics one month prior to participation, undertaking male hormone/androgen treatments (e.g., testosterone therapy, aromatase inhibitors), or following a special diet (e.g., vegetarian, vegan, low-carbohydrate).

Out of 54 potential participants who were screened remotely, 50 were eligible. Eligible participants attended a clinical trial facility for a single session in an overnight fasted state. Anthropometric measures were recorded once (height, weight, waist circumference, body mass index (BMI)). The Baecke Physical Activity Questionnaire (BPAQ), Pittsburgh Sleep Quality Index (PSQI), and irritable bowel syndrome (IBS) Severity Scoring System (IBS-SSS) were administered as covariates to control for habitual physical activity, sleep patterns, and IBS symptom severity, respectively [34–36].

A Lactulose/Rhamnose (L/R) dual sugar solution was used to determine the presence of IP. Dual sugar probe analyses are the gold standard of IP in current literature. Participants provided a 20 mL baseline blood sample via venepuncture before ingesting a dual sugar solution of 100 mL water, 5 g lactulose (Alphapharm, Sydney, Australia) and 1 g of

L-Rhamnose (Sigma-Aldrich, Merck, St. Louis, MO, USA). Participants remained at the clinic in a rested state for 1.5 h following solution ingestion, where one final 20 mL blood sample was collected. Physical activity and water intake was restricted between blood samples which were taken between 0730 h and 1230 h. Whole blood was centrifuged within 1 hr of collection and plasma and serum were stored at −80 °C until analysis.

Serum *H. pylori* Immunoglobulin G (IgG) antibodies were determined by a solid phase, chemiluminescent immunometric assay (CLIA) (Siemens Healthcare Diagnostics, Marburg, Germany) with a reportable range of 0.4 to 7.0 U/mL and an intermediate precision of 2.4% at 0.68 U/mL.

A dual sugar lactulose/rhamnose (L/R) ratio test was used as the primary measure of small IP [37,38] and were analysed using gas chromatography-mass spectrometry (GC-MS) [39–41] Based on the dose of 5 g lactulose and 0.5 g rhamnose probes administered to participants and a 1.5 h transit time to the SI, the following formula was used to calculate sugar probe recovery, expressed as a percentage and plasma sugar ratio [21,39,42]:

(1)    Recovery of lactulose (%) = plasma lactulose concentration/5 g of administered lactulose × 100

(2)    Recovery of rhamnose (%) = plasma rhamnose concentration/0.5 g of administered rhamnose × 100

(3)    L/R ratio = plasma concentration of lactulose large probe/plasma concentration of rhamnose

Zonulin, a commonly used biomarker to reflect small IP, was used as a secondary measure of IP. This was facilitated by the Adelaide Research Assay Facility using sandwich ELISA (Cusabio, Wuhan, China) read at 450 nm on a Biotek Synergy H1 plate reader (Winooski, VT, USA). Intra-assay variation was 9.3% and the lower limit of quantitation was 2.5 ng/mL. For the interpretation of zonulin results, only 32 readable values out of 50 participants were reported. Recollection of the 18 unreadable samples was unfeasible.

LBP levels as the primary measure of inflammation were determined via sandwich ELISA (Hycult Biotech, Uden, The Netherlands) with a measurable concentration range of 4.4–50 μg/mL, intra-assay precision of 2.9% at 5.7 μg/mL, and inter-assay precision of 1.4% at 5.9 μg/mL. Absorbance was read at 450 nm (Thermo Fisher Scientific, Waltham, MA, USA).

Serum high sensitivity C-reactive protein (hsCRP) levels as a secondary measure of inflammation were obtained through an immunoturbidimetric assay (ITA) (Roche Diagnostics, Rotkreuz, Switzerland) with turbidity read at 546 nm (Ortho Clinical Diagnostics, Raritan, NJ, USA). The measurable range was 0.15–20.0 mg/mL, with an intra-assay variation of 4.0% at 3.44 mg/L and an inter-assay variation was 6.8% at 3.06 mg/L concentration.

For endocrine assessment, serum T was determined using ELISA (Roche Diagnostics, Basel, Switzerland) with a working range of 0.087, 52.0 nmol/L (0.025, 15.0 ng/mL) and coefficient of variation of 19.1% at 0.31 nmol/L. Serum LH was determined using ELISA (Roche Diagnostics, Basel, Switzerland) with a re-portable range of 0.100–200 mIU/mL and an intermediate precision of 2.0% at 5.81 mIU/mL. The analytical measurement range of the serum FSH ELISA (Roche Diagnostics) was 0.100–200 mIU/mL with an intermediate precision of 3.6% for 5.33 mIU/mL.

The primary measure of interest was the correlation between *H. pylori* antibodies and IP (L/R ratio, zonulin, IBS-SSS) proportional to inflammatory (LBP, hsCRP) hypogonadism (T). One-way Analysis of Variance (ANOVA) was conducted to compare T tertiles and *H. pylori* titers with appropriate dependent variables. Pearson's correlation coefficients were used to measure associations between *H. pylori* antibodies, LBP, hsCRP, L/R ratio and IBS-SSS scores against hormones. While adjusting for age, BMI, sleep quality, and physical activity as potential confounders, multivariate analysis was undertaken to examine associations between zonulin, IP, and T levels.

Statistical analysis was performed using Statistical Package for the Social Sciences (SPSS) version 26 (IBM, Armonk, New York, NY, USA). Unless stated otherwise, data

are presented as mean $\pm$ standard deviation where $p < 0.05$ is considered statistically significant. All data were assessed for normality using Shapiro–Wilk's test where $p \leq 0.05$ was considered skewed. Abnormally distributed data were log-transformed or analysed using non-parametric tests.

## 3. Results

### 3.1. Demographics

The average age of participants was 36 years, where the majority were of normal weight (46%, 18.5–24.9 kg/m$^2$, 38% were overweight (25.0–29.9 kg/m$^2$), and 16% were obese (>30.0 kg/m$^2$) (Table 1).

**Table 1.** Anthropometric, serological, endocrine, and questionnaire data of male participants, including clinical reference ranges for serology and endocrine biomarkers ($n$ = 50).

| Data Type | n | Variable | Mean | $\pm$ | SD | Cohort Range | Reference Range |
|---|---|---|---|---|---|---|---|
| Anthropometric | 50 | Age (years) | 36.2 | $\pm$ | 5.5 | 18.0–48.0 | - |
| | 50 | Height (cm) | 174.6 | $\pm$ | 7.8 | 139.0–200.0 | - |
| | 50 | Weight (kg) | 78.8 | $\pm$ | 15.2 | 49.0–205.0 | - |
| | 50 | BMI (kg/m$^2$) | 25.6 | $\pm$ | 4.5 | 19.1–40.0 | Underweight: 16.0–18.4; Normal weight: 18.5–24.9; Overweight: 25.0–29.9; Moderately Obese: 30.0–34.9; Severely Obese: 35.0–39.9; Morbidly Obese: $\geq$40.0 [43]). |
| | 50 | Waist circumference (cm) | 85.5 | $\pm$ | 13.5 | 63.1–130.00 | >102 cm in men: increased cardiometabolic risk [44]. |
| | 50 | Physical Activity (score) | 7.84 | $\pm$ | 1.74 | 4.3–11.9 | No published reference range, however, minimum–maximum scores range between 1–15 [34]. |
| | 50 | Sleep Quality (score) | 4.14 | $\pm$ | 2.18 | 1.0–12.0 | No published reference range, however, minimum–maximum scores range between 0–21 [35]. |
| | 50 | IBS symptoms (score) | 44.23 | $\pm$ | 24.37 | 0.9–2.1 | 75–175 mild; 175–300 moderate; >300 severe cases [36]. |
| Serological | 50 | *H. pylori* igG antibodies (U/mL) | 0.98 | $\pm$ | 1.79 | 0.0–7.40 | <0.9 negative; 0.9–1.1 equivocal; >1.1 positive [45]. |
| | 50 | L/R (mmol/L) | 1.16 | $\pm$ | 1.66 | 0.0–7.9 | Diagnostic data unpublished. |
| | 38 | Zonulin (ng/mL) | 12.23 | $\pm$ | 10.81 | 0.37–41.20 | 40.0 Cusabio Human Zonulin ELISA Kit upper limit of the standard (CUSABIO TECHNOLOGY LLC, Wuhan, China). |
| | 50 | LBP (mcg/mL) | 19.90 | $\pm$ | 22.57 | 0.04–98.4 | 18.1 control; 40.0–60.0 septic shock [46]. |
| | 50 | hsCRP (mg/L) | 1.01 | $\pm$ | 1.08 | 0.40–5.1 | 1.00–3.00 slightly increased relative risk of CHD [47,48]. |
| Endocrine | 50 | T (nmol/L) | 19.18 | $\pm$ | 6.69 | 6.9–36.1 | <6.9 hypogonadism; 6.9–11.1 equivocal; >11.1 normal [1] |
| | 50 | LH (IU/L) | 6.43 | $\pm$ | 2.87 | 2.12–13.17 | 1.8–8.6 normal [3]. |
| | 50 | FSH (IU/L) | 4.54 | $\pm$ | 3.31 | 0.94–20.13 | 1.0–5.0 normal [3]. |

### 3.2. Questionnaires

#### 3.2.1. Physical Activity

The BPAQ consisted of 16 self-reported items across 3 indexes—the Work Index, the Sport Index, and Leisure-Time index. Global scores were calculated as per protocol where yielded scores ranged between the minimum–maximum values of 1–15 [34]. Cohort scores ranged between 4.3 and 11.9, with a mean of 44.23 (SD = 24.37). Higher scores indicated greater levels of physical activity relative to lower scores.

#### 3.2.2. Sleep Quality

The PSQI measured self-reported sleep quality over the 1-month interval preceding administration. Composite scores ranged between 0 and 21 and were obtained as per protocol across 19 items which formed seven components, sleep quality, latency, duration, efficiency, disturbances, use of sleeping medication, and daytime [35]. The range of cohort

scores fell between 1.0 and 12.0, with a mean of 7.84 (SD = 2.18). Higher scores indicate worse sleep quality relative to lower scores.

### 3.2.3. Irritable Bowel Syndrome

The IBS-SSS measured self-reported disease-severity as defined by physiological GI and extraintestinal (psychological/psychosomatic) symptomatology, health-related behaviours, and quality of life over the 10-day interval preceding administration. Symptoms measured included: abdominal pain and distension; stool frequency and consistency and satisfaction with bowel habits. Minimum and maximum scores of the scale range between 0 and 500 where higher scores indicate greater symptom severity. Scores between 75 and 175 are considered mild cases, whereas scores of 175–300 and >300 indicate moderate and severe cases [36,49]. While no scores from this cohort exceeded the IBS-SSS cut-off score for moderate, or severe IBS, a weak inverse relationship between IBS-SSS score and LBP was revealed (r = −0.457, *p* = 0.004).

### 3.3. Serology

#### 3.3.1. hsCRP

The mean level of hsCRP was 1.01 mg/L. This falls below the cut-off value of 1.08 mg/L used as an incidence predictor of Coronary Heart Disease (CHD) within the general population; however, is within the range of 1.00–3.00 mg/L for potentially representing a slightly increased/moderate relative risk of future CHD development [47,48]. See Table 1. As expected, both inflammatory markers hsCRP and LBP were positively correlated (r = 0.11, *p* = 0.01). This was independent of physical activity level and sleep quality. Furthermore, no correlation was revealed between hsCRP and either marker of intestinal permeability-zonulin or L/R ratio. See Figures 1 and 2. This suggests that the source of inflammation within this population does not originate from small intestinal pathology, but rather adiposity.

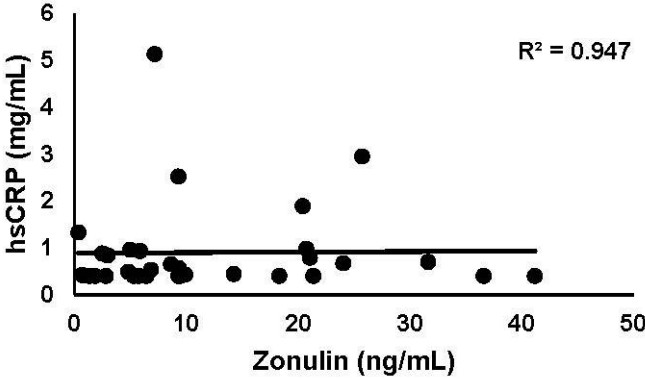

**Figure 1.** No significant relationship was revealed between hsCRP (mg/mL) and Zonulin (ng/mL).

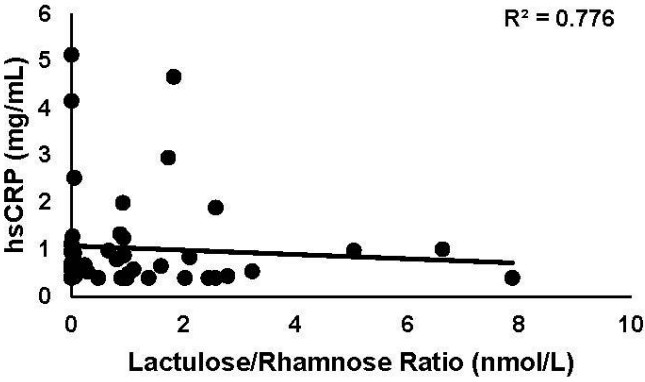

**Figure 2.** No significant relationship was revealed between hsCRP (mg/mL) and L/R ratio (nmol/L).

### 3.3.2. LBP

Average cohort LBP levels were normal, at 19.90 ng/mL (Table 1). This is well below the 20,000 ng/mL upper limit of healthy individuals and the 200,000 ng/mL value seen in acute phase immune responses like bacterial sepsis [50]. There was no significant relationship observed between LBP and T, despite wide individual variations (Figure 3).

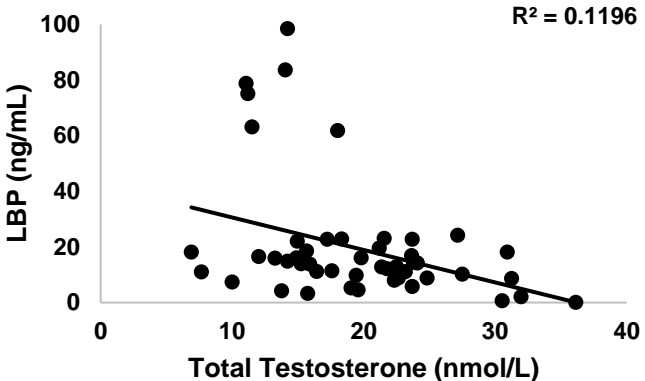

**Figure 3.** Inverse relationship between serum endotoxin (LBP) (ng/mL) and serum T (nmol/L).

### 3.3.3. *H. pylori*

As outlined in Table 1, 82% of participants were *H. pylori* negative (<0.9 U/mL), while the remaining 18% of participants were equivocal (0.9–1.1 U/mL). No participants in this cohort were considered *H. pylori* positive (>1.1 U/mL), that is, the entire study population was absent of active *H. pylori* infection [45]. *H. pylori* class IgG antibodies did not correlate with any measured variable.

### 3.3.4. Intestinal Permeability

The mean cohort plasma L/R ratio was 1.16 nmol/mL with a range of 0–7.9 nmol/mL, indicating no increased small IP (diagnostic cut-off value unpublished). There was no correlation between L/R ratio and zonulin ($p = 0.44$, $r = -0.14$). There was no correlation between L/R ratio and zonulin ($p = 0.44$, $r = -0.14$), nor L/R and other GI function markers, including *H. pylori* ($p = 0.54$, $r = 0.89$) and IBS-SSS ($p = 0.09$, $r = -0.01$). Similarly, no correlation was observed between L/R ratio and inflammatory markers, including hsCRP ($p = 0.08$, $r = -0.02$) and LBP ($p = 0.09$, $r = 0.006$). There was also no relationship between L/R ratio and T ($p = 0.017$, $r = 0.33$) (Figure 4) or other endocrine markers, including LH ($p = 0.64$, $r = 0.06$) and FSH ($p = 0.52$, $r = -0.09$).

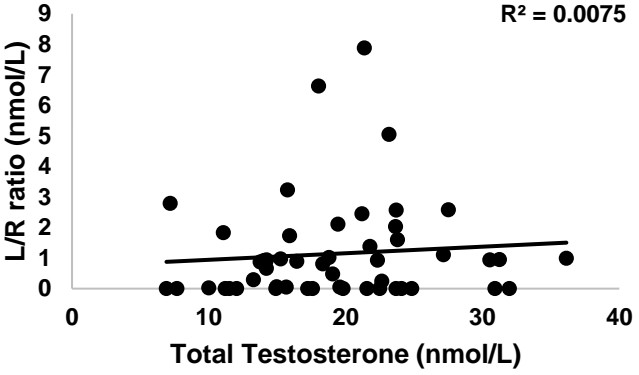

**Figure 4.** Relationship between L/R ratio (nmol/L) and serum T (nmol/L).

### 3.4. Endocrinology

Mean T at 19.19 nmol/L (SD = 6.69) (range 6.90–36.1 nmol/L) fell within the normal clinical range. Therefore, no participants were considered to have HG (population range

6.90–36.1 nmol/L) [51]. Mean LH and FSH levels fell within normal clinical ranges, respectively, at 6.43 IU/L (SD = 2.87) and 4.53 IU/L (SD = 3.31) [52]. As expected, LH showed a significant association with T (r = 0.33, p = 0.01) and FSH (r = 0.46, p = 0.001). A significant inverse agreement between T and hsCRP (r = −0.398, p = 0.004), independent of physical activity level and sleep quality, was revealed, as seen in Figure 5.

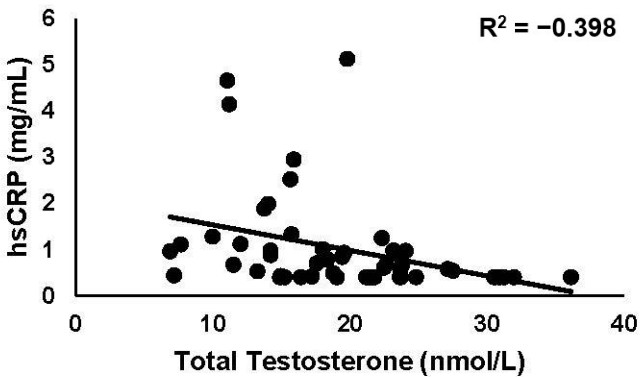

**Figure 5.** Inverse relationship between hsCRP (mg/mL) and serum T (nmol/L).

## 4. Discussion

LPS exposure-induced inflammation can impair testicular function. We aimed to explore the potential relationship between gastric and small intestinal pathology, namely *H. pylori* and IP which have both been shown to increase LBP concentrations- and consequentially, ME on testicular function in men.

No participants in this cohort displayed *H. pylori* infections. In lieu, no correlation between *H. pylori* and IP, ME, or T was observed. This may be because *H. pylori*, despite its classification as a Gram-negative bacterium, has a markedly lower immunogenic response compared to *E. coli*, for example, due to differences in the molecular structure of LPS [53,54]. As such, circulatory *H. pylori* endotoxins may potentially elicit a smaller systemic inflammatory response compared to cellular damage occurring locally within the gastric mucosa. Furthermore, compared to other strains, aggressive CagA+ variants of *H. pylori* display higher virulence through increased biosynthesis of LPS within the stomach. This contributes to increased severity of gastric complications including increased mucosal concentrations of inflammatory cytokines in addition to increased extra-gastric complication severity, including higher circulatory endotoxin load, compared with other variants [55,56]. These gastric and extra-gastric conditions were not measured in this study and may not be entirely reflective of the extent of *H. pylori* virulence experienced in this cohort. Additionally, one limitation of this study was the measure of *H. pylori* used (CLIA analysis of IgG antibodies) which differed from the gold-standard method of urea breath testing, and produced results that were not clinically meaningful. Serological analysis of serum antibody titres is not considered a valid diagnostic method for *H. pylori* determination due to its inability to differentiate between past and present *H. pylori* infection [57,58].

Despite no evidence of moderate or severe IBS within this cohort, the inverse relationship between IBS-SSS scores and LBP was expected and in line with literature revealing persistent, low-grade inflammation at both a systemic, and local, mucosal level observed in IBS patients [59]. Similarly, our results revealed a significant, inverse relationship between hsCRP and testosterone (r = −0.398, p = 0.004). Significance was maintained by controlling for covariates of physical activity level and sleep quality; however, these associations did not hold true when BMI was controlled for. This reflects existing literature showing links between increased BMI, inflammation, and androgen deficiency [7,8]. It also must be noted that one drawback of this study was the lack of controlling for Inflammatory Bowel Disease (IBD). Both IBD and IBS mucosal biopsies are associated with increased IP, and both conditions share symptoms typical of IBS. Therefore, this study was unable to

differentiate IBS symptom presence and severity from IBS or IBD, both indicated decreased barrier function which precedes ME and subsequent androgen deficiency [59,60].

No evidence of small IP was present within this cohort despite a methodological approach appropriate for small IP determination. While the vast majority of literature utilized dual-sugar urinary excretion analysis involving transit times ranging between 5 h and overnight for IP measurement, serum L/R methodological approaches where sample collection occurs between 1 and 1.5 h are more appropriate for small IP determination in addition to boasting higher sensitivity [37,38,61].

While SIBO is known to increase small IP, it was not measured because the collection of bacterial culture ($>10^4$ bacteria per ml of jejunal aspirate) via endoscopy was not feasible and is, therefore, a limitation of this study.

The lack of significant correlation between T levels and both *H. pylori* and IP suggests that a colonic source of circulatory endotoxin responsible for ME-induced reductions in T, if present, may be more likely. Average healthy men contain a colonic bacterial load exceeding that of the stomach and SI by $10^{12}$ fold, which supports this theory [20,27]. Additionally, findings of a recently conducted study by Tremellen et al., 2023 [62] highlighted no significant link between proton pump inhibitor (PPI) and antacid intake to T concentration, despite both being known risk factors for SIBO and IP–a finding that reinforces the above theory.

There was no correlation between zonulin and LBP, nor zonulin and T levels. These results are discrepant with a recent publication showing significant positive associations between zonulin and T levels and LBP, independent of adiposity, suggesting a potential link between small IP and HG [63]. A limitation and potential reason underlying this intra-study discrepancy was the use of ELISA-determined zonulin, which is of debatable validity. Preliminary findings have reported that ELISA-determined serum zonulin levels may incorrectly reflect non-zonulin immune factors, such as intestinal lumen complement [64,65]. Furthermore, research by Pearce, Hill, and Tremellen [63] included participants with HG, whereas this cohort had no participants with HG. Additionally, the zonulin concentration values of 18 participants in this cohort were unintelligible, reducing the statistical power of this study. Considering that Metformin, a glucose-mediating medication used to manage type II diabetes, has been shown to augment intestinal microbial composition and T levels in women with PCOS, differences in the study population mean BMI (30.2 versus 25.6), and thereby glucose uptake and insulin-sensitivity (unmeasured in both studies), may further explain this discrepancy [66,67]. Supporting this notion of metabolic stress, namely inflammatory mechanical and metabolic stress-states occurring with high-intensity exercise and long periods of exercise, have been shown to increase intestinal permeability (as measured by zonulin) via alterations to intestinal microflora and reductions of gastrointestinal blood flow. This can cause mucosal ischemia (decreased blood flow and oxygen within the intestinal epithelial barrier), which results in villous injury and subsequent losses in barrier function via decreased expression of tight junction proteins. However, most of the participants within this cohort were sedentary/recording minimal levels of exercise, meaning that any analysis of the extent of physical activity levels, *H. pylori*, and IP would have been underpowered. This study was unable to determine the extent of confounding factors in the association between *H. pylori*, IP, and T levels due to underpowered analysis. Larger studies are needed to examine the effect of these potential confounders on the outcome variables.

## 5. Conclusions

In conclusion, gastric pathology marker *H. pylori* IgG antibodies, GI pathology marker IBS symptom severity, and small IP markers zonulin and L/R ratio, were not found to influence ME or HG in this cohort. This is likely a result of the entire cohort falling within healthy clinical ranges for *H. pylori*, IP, and T. It was not possible to observe the influence of gastric and SI pathologies on testicular function in the absence of gastric/GI/testicular disorder. Contributing factors of these results include reduced statistical power due to 18 participants with unreadable zonulin values where recollection was unfeasible, poten-

tially lower immunogenic response of *H. pylori* strains present in this cohort, questionable validity of *H. pylori* and IP determination, absence of colonic pathology/colonic IP, underpowered analysis of confounding variables, and lack of SIBO determination. Future investigations are required to elucidate more precisely the gastrointestinal, and particularly colonic factors, contributing to ME-induced HG, ideally utilising diagnostic methodologies used in clinical practice.

**Author Contributions:** Conceptualization, L.N.P., K.L.P. and K.P.T.; Data curation, L.N.P., K.L.P. and K.P.T.; Formal analysis, L.N.P., K.L.P. and K.P.T.; Funding acquisition, K.P.T.; Investigation, L.N.P., K.L.P., C.D.T. and K.P.T.; Methodology, L.N.P., K.L.P., C.D.T. and K.P.T.; Project administration, K.J.M., K.L.P. and K.P.T.; Resources, K.L.P., C.D.T. and K.P.T.; Supervision, K.J.M., K.L.P. and K.P.T.; Validation, L.N.P., K.L.P. and K.P.T.; Visualisation, L.N.P. and K.L.P.; Writing—original draft, L.N.P., K.J.M., K.L.P. and K.P.T.; Writing—review and editing, L.N.P., K.J.M., C.D.T. and K.P.T. All authors have read and agreed to the published version of the manuscript.

**Funding:** This research was funded by Repromed.

**Institutional Review Board Statement:** The study was conducted in accordance with the Declaration of Helsinki and approved by the University of South Australia Human Research Ethics Committee (#202371, 9 October 2019).

**Informed Consent Statement:** Informed consent was obtained from all participants involved in the study.

**Data Availability Statement:** Not applicable.

**Conflicts of Interest:** The authors declare no conflict of interest.

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
