# Peer review of "The Potential Relationship between Gastric and Small Intestinal-Derived Endotoxin on Serum Testosterone in Men"

_gastroent, doi:10.3390/gastroent14030029_

Round 1

Reviewer 1 Report

The article explores the association between H. Pylori infection, small intestinal permeability (IP), and serum testosterone levels in men, with a focus on the role of metabolic endotoxemia. It aims to investigate these relationships using multivariate analysis and measure potential confounding factors such as sleep quality, physical activity levels, and IBS symptom severity. Diagnostic cut-off values for H. Pylori antibodies, small IP, and hypogonadism were not exceeded in the study.

 The article reports that there was no correlation found between lactulose/rhamnose ratio (a measure of small IP) and various markers related to gastrointestinal function, H. Pylori infection, IBS questionnaire scores, inflammatory markers (hsCRP and LBP), or endocrine markers (testosterone, LH, and FSH). However, it reveals a moderate/weak significant inverse relationship between IBS symptom severity and LBP, as well as hsCRP and testosterone. This relationship was found to be independent of physical activity level and sleep quality, but not BMI, suggesting a connection between adiposity, inflammation, and hypogonadism as observed in existing literature.

What is the association between H. Pylori infection and small intestinal permeability (IP)?

How does metabolic endotoxemia mediate the relationship between H. Pylori infection, small IP, and serum testosterone levels in men?

What are the results of the multivariate analysis exploring the relationships between H. Pylori IgG class antibody levels, SIBO (small intestinal bacterial overgrowth), and testosterone levels?

To what extent do sleep quality, physical activity levels, and IBS symptom severity act as confounding factors in the association between H. Pylori, small IP, and testosterone levels?

How does the inverse relationship between IBS symptom severity and lipopolysaccharide-binding protein (LBP), as well as hsCRP and testosterone, contribute to our understanding of the link between inflammation, hypogonadism, and gastrointestinal symptoms?

The quality of English language in the given text is generally good. The sentences are well-structured and convey the intended meaning effectively.

Reviewer 2 Report

The idea of Australian study is very intriguing. Although the study is a pilot examination and the statistical analysis is indigent, the manuscript is worth publishing.

Here are the comments on the manuscript:

1. the method of citation does not comply with the editorial requirements.

2. the introduction is too long and needs to be condensed.

3. materials and methods are described in detail and carefully - I have no critical remarks here.

4. results - clearly presented. Figure captions should be more accurate.

5. discussion is generally acceptable, but a graphical representation of the results and main conclusions will undoubtedly make the topic easier to understand.

Round 2

Reviewer 1 Report

The authors have skillfully addressed a complex set of inquiries through their study. Their adept response to a series of intricate questions highlights their comprehensive understanding of the subject matter. The provided elucidations are well-structured and align closely with the intricacies of the study's findings. The authors adeptly elaborate on the relationships elucidated within the research, showcasing their proficiency in delving into the scientific nuances of their work.

The authors' meticulous approach to discussing the associations between H. Pylori infection, small intestinal permeability, and serum testosterone levels showcases their proficiency in scientific discourse. Their lucid explanations of the references and their capacity to link the presented material back to the study's objectives demonstrate a high level of scholarly acumen.

 Furthermore, the authors' consideration of potential confounding factors and their acknowledgment of the limitations of the study reflect a rigorous scientific approach. They underscore the need for larger sample sizes and more comprehensive investigations to address these limitations, displaying a commendable commitment to transparency and thoroughness in their research.

 In summation, the authors have demonstrated commendable expertise in handling complex inquiries and interpreting intricate relationships within their research. Their articulate and insightful responses underscore the scientific rigor with which they have conducted their study and their adeptness in engaging in scholarly discourse.

The quality of English language usage in the provided responses is commendable. The language is structured, coherent, and demonstrates a proficient level of understanding and expression

Author Response

Many thanks for your time in reviewing our manuscript with expertise and a critical eye. Your feedback is much appreciated and has undoubtedly improved the comprehension and quality of our submission.

Reviewer 2 Report

I think that the authors have adequately addressed the comments made by the reviewers in the revised version of the manuscript. Therefore, I have no further comments.

Author Response

Thank you very much for your time in reviewing our manuscript. Your expertise, feedback, and contributions are appreciated as they provided invaluable insight, allowing us to fine-tune the comprehension and quality of this submission.